# Prediction of concrete strength using response surface function modified depth neural network

**Xiaohong Chen**[1]*, **Yueyue Zhang**[1], **Pei Ge**[2]

**1** Railway Engineering College, Zhengzhou Railway Vocational & Technical College, Zhengzhou, China,
**2** School of Civil Engineering and Architecture, Changzhou Institute of Technology, Changzhou, China

* 954692912@qq.com

## Abstract

In order to overcome the discreteness of input data and training data in deep neural network (DNN), the multivariable response surface function was used to revise input data and training data in this paper. The loss function based on the data on the response surface was derived, DNN based on multivariable response surface function (MRSF-DNN) was established. MRSF-DNN model of recycled brick aggregate concrete compressive strength was established, in which coarse aggregate volume content, fine aggregate volume content and water cement ratio are influencing factors. Furthermore, the predictive analysis and extended analysis of MRSF-DNN model were carried out. The results show that: MRSF-DNN model had high prediction accuracy, the correlation coefficient between the real values and the forecast values was 0.9882, the relative error was between -0.5% and 1%. Furthermore, MRSF-DNN had more stable prediction ability and stronger generalization ability than DNN.

## 1 Introduction

An important index of concrete mechanical properties is concrete compressive strength (CCS), moreover, the first step of concrete mix proportion design plan is the calculation for CCS. Various studies have been made contributions to the prediction model for CCS. Meng et al. [1] presented that the CCS was most affected by water cement ratio (W/C) and was linear with the decrease of W/C. Nakata et al. [2] further pointed out that concrete strength was mainly affected by W/C and compactness. In literatures [3–5], among the factors influencing CCS, W/C and the volume content of coarse aggregate (CA) were the most important. In addition, Guo et al. [6] reported that CCS not only showed a linear decreasing connection with W/C, but also a nonlinear connection with CA properties.

In terms of the calculation method of CCS, Zhang et al. [7] proposed a prediction model of recycled aggregate concrete (RAC) compressive strength on the basis of considering W/C and the ratio of coarse to fine aggregate (FA). What is more, Chen et al. [8] developed a model for RAC compressive strength with concrete curing age and the ratio of recycled brick aggregate (RBA) replacing natural aggregate (NA) as main parameters. Zhang et al. [9] revised Bolomey formula by considering the specific gravity, bulk density and water absorption of recycled

**Competing interests:** The authors declare that they have no known competing financial interests or personal connections that could have appeared to influence the work reported in this paper. This does not alter our adherence to PLOS ONE policies on sharing data and materials.

aggregate (RA). Fang et al. [10] found that there was a parabolic connection between the strength of RAC and the crushing index of RA. They used the crushing index of RA as a parameter to regress the calculation formula of the strength of RAC. Younis and Pilakoutas [11] found that CCS was affected not only by W/C, but also by the properties of CA. So they established linear and nonlinear prediction models among CCS, W/C and CA properties respectively. In another study, Xiong et al. [12] pointed out that in addition to W/C, the factors affecting CCS include the cement paste strength, the bonding strength of interface transition zone and the volume content of CA. Furthermore, the prediction model for CCS was also established considering these factors.

The above researches show that concrete strength is not only related to W/C, but also related to cement mortar strength, interface bond strength, CA properties, compactness and other factors. The links between one predict result and multiple variables can be expressed by multivariate linear or nonlinear functions. However, the establishment of different multivariate functions needs to be based on different independent variable (IV), which has a large amount of calculation and is inconvenient to use. With the development of numerical algorithms, the method of artificial neural network (ANN) has emerged, that can quickly learn the data characteristics according to input data and give the prediction results of multiple IVs. The prediction results show that ANN shows strong learning ability and high prediction efficiency [13]. Some scholars have established some ANN prediction models for CCS with multiple IVs. In the study by Duan et al. [14], an ANN model for predicting CCS was proposed considering different W/C and different physical properties of CA. Deng et al. [15] predicted the impact of water cement ratio, recycled coarse aggregate substitution rate, recycled fine aggregate substitution rate and fly ash substitution rate on the compressive strength of recycled aggregate concrete through convolutional neural network. The results show that compared with traditional neural network models, the prediction model based on deep learning has the advantages of high accuracy, high efficiency, and strong generalization ability. Chou et al. [16] proposed a convolutional neural network based on computer vision for predicting the compressive strength of ready-mixed concrete. The results show that the neural network based on computer vision is superior to the neural network based on numerical data in all evaluation indicators. Xu et al. [17] proposed an ANN model and a multivariate nonlinear model, which can predict CCS. It displayed that the accuracy of these two models was all higher than that of the existing prediction models for CCS. According to Deshpande et al. [18], the forecast correctness of multivariate nonlinear model and tree model was lower than that of ANN model. That was because multivariate nonlinear model and tree model cannot revise the forecast results according to the new input data. In the study of Khademi and Behfarnia [19], the multivariate linear model and BP neural network (BPNN) model for CCS were established respectively. It displayed that the multivariate linear model showed lower prediction correctness and efficiency than the BPNN model. In addition, the BPNN would get different accuracy prediction results by setting different calculation steps each time. Furthermore, Tu et al. [20] used genetic algorithm (GA) to optimize the prediction accuracy and stability of BPNN. The result displayed that the optimized BPNN model by GA showed better prediction accuracy and stability.

Response surface methodology has been widely used in concrete performance analysis, optimization and prediction. Application of response surface method in concrete performance analysis, Shi et al. [21] used response surface methodology to analyze the effects of silane rubber and nano $SiO_2$ on the pore structure and mechanical properties of concrete. Adamu et al. [22] used response surface methodology to optimize the effects of the amounts of plastic waste, fly ash and graphene nanotube on the strength and water absorption of concrete. The results showed that the optimal mixing amounts of plastic waste, fly ash and graphene nanoplate were

15.3%, 6.07% and 0.22%, respectively. Hua et al. [23] used the response surface method (RSM) to analyze the influence of Seawater/Potassium Silicate, Potassium Hydroxide/Potassium Chloride, Sodium Laureth Ether Sulfate/Benzalkonium Chloride and Hydrogen Peroxide/ Nanocellulose on the density of geopolymer foam concrete. Adamu et al. [24] used response surface methodology to analyze the impact of calcium carbide waste and rice husk ash on the water absorption and permeability of concrete. The results showed that both calcium carbide waste and rice husk ash had a negative impact on the durability of concrete, but rice husk ash had a greater negative impact. Tunc et al. [25] used the response surface method to analyze the effects of water cement ratio, aggregate/cement and Los Angeles abrasion rate on the compressive strength and splitting tensile strength of concrete. Ferdosian et al. [26] used response surface methodology to study the effects of silica fume, ultra-fine fly ash and sand as three main constituents on the workability and compressive strength of ultra-high performance concrete. Application of response surface method in multiobjective optimization, Luo et al. [27] provided a method for optimizing the mix ratio of dune sand concrete based on dune sand/fine aggregate, basalt fiber content, water/ cement and sand/aggregate. The results show that the model established by response surface method is effective and can accurately predict the performance of dune sand concrete. Hamada et al. [28] used response surface methodology to study the effects of nano palm oil fuel ash and palm oil clinker partially replacing cement (0, 15% and 30%) and coarse aggregate (0, 50% and 100%) on the workability and compressive strength of concrete. The results show that response surface methodology has achieved satisfactory results in optimizing the amount of nano palm oil fuel ash and palm oil clinker. When containing 0% palm oil clinker and 15% nano palm oil fuel ash, the compressive strength of concrete is the maximum; When containing 100% palm oil clinker and 30% nano palm oil fuel ash, the compressive strength of concrete is the lowest. Amiri et al. [29] studied the effects of water cement ratio, cement content, gravel content and coal gangue content on the compressive strength and water absorption of concrete. A combination of response surface method and expected function method was used to optimize the relationship between independent variables and response variables for six schemes. Kursuncu et al. [30] used the response surface method and artificial neural network methods to study the effects of waste marble powder and rice husk ash partially replacing fine aggregate and cement on the compressive strength, flexural strength, porosity and thermal conductivity of foam concrete. Zhang et al. [31] designed a Box-Behnken model using the response surface method to study the effects of different amounts of silica fume, fly ash and carbon fiber on the compressive strength and strain sensitivity coefficient of sprayed reactive powder concrete. Siamardi et al. [32] established a model using the response surface method to predict the workability and hardening performance of powder based lightweight self-compacting concrete produced by partially replacing normal weight aggregate with coarse grained lightweight expansive clay aggregate. Shahmansouri et al. [33] used response surface methodology to study the effects of sodium hydroxide concentration, natural zeolite and silica fume on the mechanical properties of geopolymer concrete and obtained the optimal design variables. Application of response surface method in prediction of concrete properties, Hammoudi et al. [34] used response surface method and artificial neural network methods to predict the compressive strength of recycled coarse aggregate concrete. The statistical results show that both response surface method and artificial neural network method are powerful tools for predicting compressive strength. However, the artificial neural network model shows better accuracy. Awolusi et al. [35] used response surface methodology to predict and optimize the impact of limestone powder and steel fiber content on the workability and hardening performance of concrete. The results show that the response surface method has high accuracy in predicting the compressive strength, splitting tensile strength, slump and water absorption of concrete. Gupta et al. [36] established a statistical model for

predicting the compressive strength of concrete using the response surface method. The results show that the prediction error of the response surface model is about 3.63%. Application of response surface method in concrete mix design, Zhang et al. [37] used response surface methodology to optimize the ratio among the ideal paste thickness, actual paste thickness and void content of recycled aggregate permeable concrete. Güneyisi et al. [38] also used response surface methodology to optimize the relationship among the three parameters of fly ash, metakaolin and cement in the mix ratio of high-performance concrete.

A new idea for prediction method of concrete strength with multiple IV is provided by ANN, but ANN can not control the discreteness of input data and revise the training data in the training process. In this paper, the multivariable response surface function was used to revise the input data and training data, the loss function considering the data on the response surface was derived. A deep neural network based on multivariate response surface function (MRSF-DNN) was established, which not only has the stability of multivariable response surface function, but also has the learning ability and generalization ability of deep neural network (DNN). At the same time, MRSF-DNN eliminates the data with large errors in the input and training process and improves the accuracy of prediction, which also improves the stability of prediction. The method of MRSF-DNN is not only used to predict the concrete strength of multiple independent variables, but also can be used to predict the mechanical properties and durability of concrete with multiple independent variables. The biggest advantage of MRSF-DNN is that the errors of input and training data which exceed σ of the response surface function can be eliminated to ensure the prediction accuracy and stability.

## 2 Deep neural network

### 2.1 The structure of deep neural network

A complete ANN includes input layer, output layer and hidden layer as shown in Fig 1. The input layer is a layer that accepts input data, the output layer is a layer that outputs results, the hidden layer is a layer for data processing. In most cases, the number of input layer and output layer is only one layer, the number of nodes can be set as required, however, the amount of layers and nodes of the hidden layer can be set freely. When the hidden layer is only one layer, it is a simple ANN as plotted in Fig 1(a); when the hidden layer is multi-layer, it is a DNN as shown in Fig 1(b).

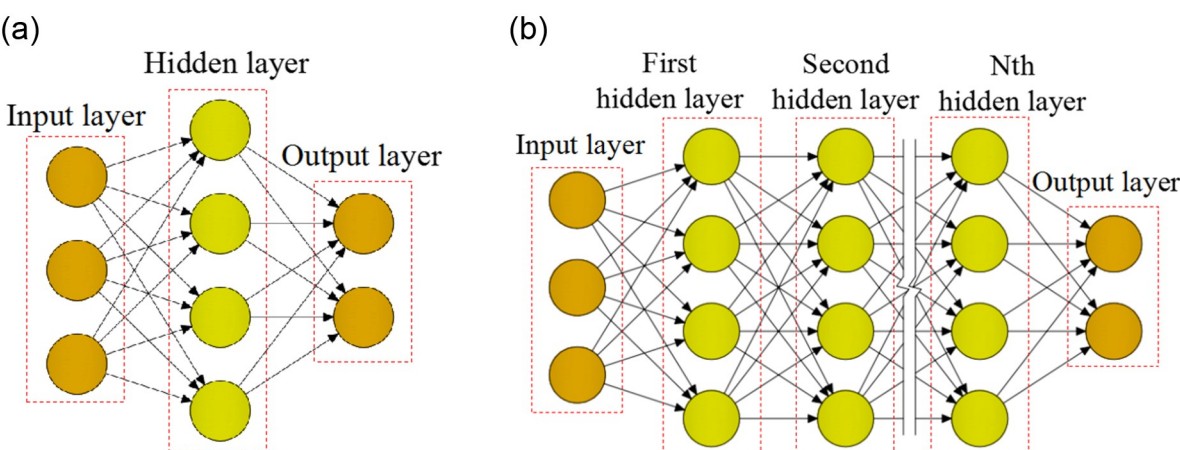

**Fig 1. The structure of ANN: (a) Simple ANN, (b) Deep neural network.**

## 2.2 The calculation principle of deep neural network

In DNN, whether it is input layer or hidden layer, the input data must be transmitted and processed through each layer. The node data of each layer will be transmitted to all nodes of the next layer, the nodes of the next layer will accept all the node data of the previous layer. The node data of the next layer will be affected by the node data of the previous layer. The weight function represents the contribution of the node data of the previous layer to the node data of the next layer. Then the node values of $1\sim j$ layers can be expressed as Eq (1):

$$\begin{cases} a_1 = w_1 \times x_1 + w_2 \times x_2 + \cdots + w_i \times x_i + b_1 \\ a_2 = w_1 \times x_1 + w_2 \times x_2 + \cdots + w_i \times x_i + b_2 \\ \qquad\qquad\qquad \cdots \\ a_j = w_1 \times x_1 + w_2 \times x_2 + \cdots + w_i \times x_i + b_j \end{cases} \tag{1}$$

Where: $a_1$, $a_2$, $\cdots$, $a_j$ are the node data of the next layer; $x_1$, $x_2$, $\cdots$, $x_i$ are the node data of the previous layer; $w_1$, $w_2$, $\cdots$, $w_i$ are weight values; $b_1$, $b_2$, $\cdots$, $b_i$ are bias terms.

In many cases, the functional equation among input data and output data is not simple linear, so an activation function is required to activate nonlinear mapping on input data. The common activation functions are *sigmoid*(), *tanh*(), *relu*(), etc. The most frequently used activation equation is *sigmoid*() as displayed in Eq (2):

$$f(z) = sigmoid(z) = \frac{1}{1 + e^{-z}} \tag{2}$$

Considering the activation function, Eq (1) can be expressed as Eq (3):

$$\begin{cases} a_1 = f(w_1 \times x_1 + w_2 \times x_2 + \cdots + w_i \times x_i + b_1) \\ a_2 = f(w_1 \times x_1 + w_2 \times x_2 + \cdots + w_i \times x_i + b_2) \\ \qquad\qquad\qquad \cdots \\ a_j = f\left(w_1 \times x_1 + w_2 \times x_2 + \cdots + w_i \times x_i + b_j\right) \end{cases} \tag{3}$$

Where: $f()$ is the activation function.

Eq (4) can be obtained by rewritten Eq (3) into matrix form:

$$a = f\left(\begin{pmatrix} w_{11} + w_{12} + \cdots + w_{1j} \\ w_{21} + w_{22} + \cdots + w_{2j} \\ \cdots \\ w_{i1} + w_{i2} + \cdots + w_{ij} \end{pmatrix} \begin{pmatrix} x_1 \\ x_2 \\ \\ x_j \end{pmatrix} + \begin{pmatrix} b_1 \\ b_2 \\ \\ b_j \end{pmatrix}\right) = f(Wx + B) \tag{4}$$

Where: $W$ is weight function matrix; $B$ is bias term vector.

The translation, transformation and spatial rotation of input data can be operated by $Wx +B$, the nonlinear mapping of input data can be operated by $f()$. The more hidden layers of DNN, the more nonlinear transformation times of the activation function to input data, the more features of input data are learned. For many complex prediction problems, multiple hidden layers are required to achieve nonlinear mapping of input data, so the learning and prediction effect of DNN is better than that of simple ANN. For multi-layer fully connected DNN,

Eq (4) can be rewritten as Eq (5):

$$
a^i = \begin{pmatrix} a^i_1 \\ a^i_2 \\ \\ a^i_n \end{pmatrix} = f\left( \begin{pmatrix} w^i_{11} + w^i_{12} + \cdots + w^i_{1m} \\ w^i_{21} + w^i_{22} + \cdots + w^i_{2m} \\ \cdots \\ w^i_{n1} + w^i_{n2} + \cdots + w^i_{nm} \end{pmatrix} \begin{pmatrix} a^{i-1}_1 \\ a^{i-1}_2 \\ \\ a^{i-1}_m \end{pmatrix} + \begin{pmatrix} b^i_1 \\ b^i_2 \\ \\ b^i_n \end{pmatrix} \right) = f(W^i a^{i-1} + b^i) \quad (5)
$$

Where: $a^i$ is the node data of $i$th layer; $n$ is the number of nodes in $i$th layer; $m$ is the number of nodes in ($i$-1)th layer; $w^i$ is the weight from ($i$-1)th layer to $i$th layer; $w^i_{nm}$ is the weight from the output of the $m$th node of the ($i$-1)th layer to the $n$th node of the $i$th layer; $b^i$ is the value of the bias term when the $i$th layer is calculated.

## 2.3 The loss function of deep neural network

The values of $w$ and $b$ in DNN are randomly generated. By continuously adjusting the values of $w$ and $b$ in DNN, the forecast value is constantly close to the real value. The deviation degree between the real value and the forecast value is defined as loss function. The common loss functions includes mean square error loss function, 0–1 loss function, absolute value loss function and so on.

0–1 loss function is shown as Eq (6). If the real value is equivalent to the forecast value, the loss value is 0; otherwise, the loss value is 1.

$$
L(y, f(x)) = \begin{cases} 1, y = f(x) \\ 0, y \neq f(x) \end{cases} \quad (6)
$$

Where: $L()$ is the loss function.

The absolute difference between the real value and the forecast value is the absolute value loss function as shown in Eq (7):

$$
L(y, f(x)) = |y - f(x)| \quad (7)
$$

The calculation formula for mean square error loss function is displayed in Eq (8):

$$
L(y, f(x)) = \frac{1}{n} \sum_{i=1}^{n} (y_i - f(x_i))^2 \quad (8)
$$

## 2.4 The gradient descent of loss function

For samples ($X$, $Y$), where $X$ and $Y$ are standard input data and output data respectively, $X$ is calculated by DNN to obtain $Y$. However, the parameters in DNN is unknown. The calculation process of DNN is to constantly update the parameters according to $X$ to minimize the loss function value. The forecast value of $X$ in the sample can be calculated by DNN, the loss value can be obtained from the forecast value of $X$ and $Y$ according to Eq (7). The loss function value can be reduced by adjusting the parameters in DNN, then the parameters in DNN can be regarded as IV, the value of loss function can be regarded as a function of dependent variable (DV).

The purpose of increasing or decreasing the value of a function can be achieved by adjusting the increase or decrease of IV. In order to make the forecast value of DNN close to the real value, the weight or bias term should move a little in the opposite direction of its gradient, where the gradient is controlled by the learning rate. If the gradient is greater than 0, the weight or bias term should be reduced a little; if the gradient is less than 0, the weight or bias term should be increased a little. The training process of DNN is to find the weight and bias

term that minimize the loss function through step-by-step iteration. The calculation method of gradient descent are Eqs (9) and (10):

$$W_{ij} = W_{ij} - \alpha \frac{\partial}{\partial W_{ij}} L(w, b) \tag{9}$$

$$b_i = b_i - \alpha \frac{\partial}{\partial b_i} L(w, b) \tag{10}$$

Where: $W_{ij}$ and $b_i$ are the parameters to be optimized in DNN; $L(w,b)$ is loss function; $\partial L(w,b)/\partial W_{ij}$ is the partial derivative of the loss function to $W_{ij}$; $\partial L(w,b)/\partial b_{ij}$ is the partial derivative of the loss function to $b_i$; $\alpha$ is the learning rate.

The input data of the sample can be iterated according to Eqs (9) and (10) to update the values of $W_{ij}$ and $b_i$ until $L(w,b)$ satisfies a certain set condition.

## 3 The multivariable response surface function

### 3.1 The multivariable linear response surface function

The main factors affecting the strength of concrete generally include two or three. If there are more than three main affecting factors, the principal factor analysis method can be used to find the most important two or three main affecting factors [39, 40]. So two or three main affecting factors are discussed in this paper. Each affecting factor is an IV, the strength of concrete is the DV. When each IV has a good linear connection with the DV, the multivariable linear response surface function can be used for regression. When there are an IV and a DV, the connection between them can be expressed as a linear function of one variable, which can be expressed as a two-dimensional graphic; when there are a DV and two IVs, the connection among them can be expressed as binary linear function, which can be expressed as a three-dimensional graphic; when there are a DV and three IVs, the connection among them can be expressed by ternary linear function, which cannot be directly expressed as a three-dimensional graphic, but can be expressed as three three-dimensional response surface graphics as shown in Fig 2.

The calculation formula of ternary linear function is Eq (11):

$$y^R = Ax_1 + Bx_2 + Cx_3 + D \tag{11}$$

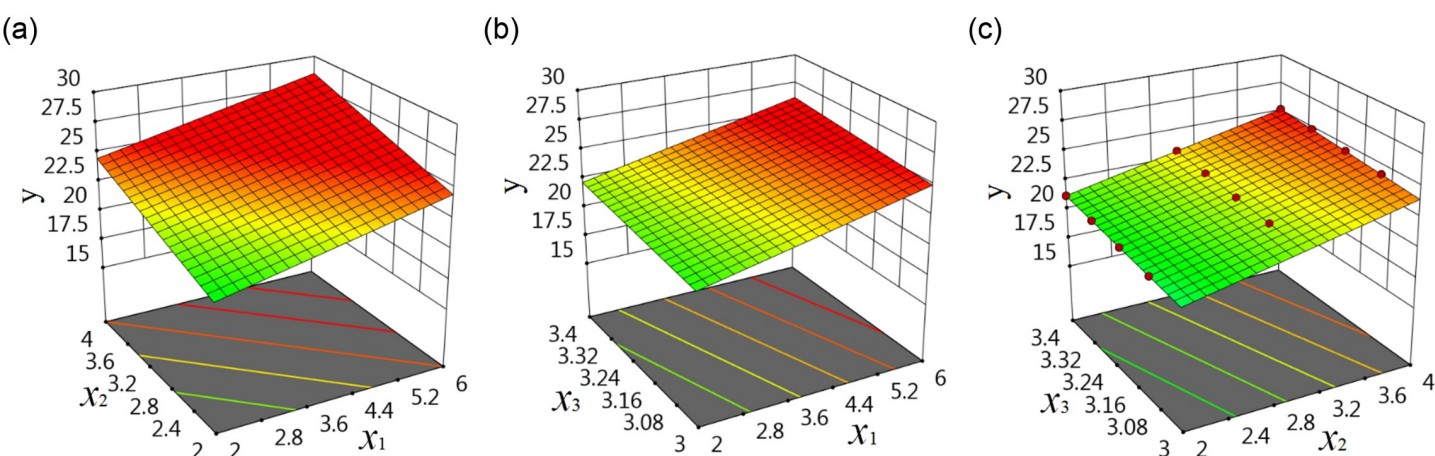

**Fig 2. Linear response surface of three IVs:** (a) The connection among $x_1$, $x_2$ and $y$, (b) The connection among $x_1$, $x_3$ and $y$, (c) The connection among $x_2$, $x_3$ and $y$.

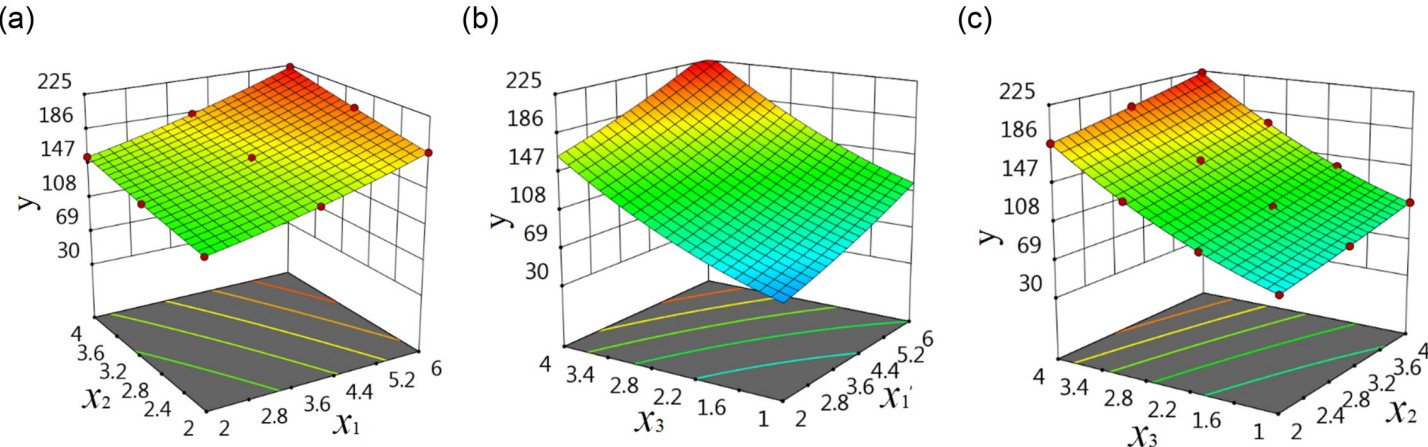

**Fig 3. Nonlinear response surface of three IVs: (a) The connection among $x_1$, $x_2$ and $y$, (b) The connection among $x_1$, $x_3$ and $y$, (c) The connection among $x_2$, $x_3$ and $y$.**

Where, $y^R$ is the response surface function; $A$ $B$, $C$ and $D$ are regression coefficients.

## 3.2 The multivariable nonlinear response surface function

When each IV has a nonlinear connection with the DV, the multivariable nonlinear response surface function is used for regression. When there are a DV and an IV, the connection between them can be expressed by a nonlinear function of one variable, which can be expressed as a two-dimensional graphic; when there are a DV and two IVs, the connection among them can be expressed as a binary nonlinear function, which can be expressed as a three-dimensional graph; when there are a DV and three IVs, the connection among them can be expressed by a three-dimensional nonlinear function, which cannot be directly expressed as a three-dimensional graphic, but can be expressed as three three-dimensional response surface graphics as shown in Fig 3.

The calculation formula of binary nonlinear function is Eq (12):

$$y^R = Ax_1 + Bx_2 + Cx_1^2 + Dx_1x_2 + Ex_2^2 + F \tag{12}$$

Where, $A$, $B$, $C$, $D$, $E$ and $F$ are regression coefficients.

The calculation formula of ternary nonlinear function is Eq (13):

$$y^R = Ax_1 + Bx_2 + Cx_3 + Dx_1x_2 + Ex_1x_3 + Fx_2x_3 + Jx_1^2 + Hx_2^2 + +Kx_3^2 + L \tag{13}$$

Where, $A$, $B$, $C$, $D$, $E$, $F$, $J$, $H$, $K$ and $L$ are regression coefficients.

## 4 The revision of DNN by multivariable response surface function

### 4.1 The revision of loss function of deep neural network by multivariable response surface function

DNN has strong learning ability, but it can not control the discreteness of input data and training data. Therefore, the multivariable response surface function is used to revise input data and training data in this paper.

The calculation formula of data expectation is Eq (14):

$$\mu = \frac{1}{N}\sum_{i=1}^{N} y_i \tag{14}$$

Where: $\mu$ is data expectation; $y_i$ is data.

The calculation formula of standard deviation of data is Eq (15):

$$\sigma = \sqrt{\frac{1}{N}\sum_{i=1}^{N}\left(y_i^R - \mu\right)^2} \tag{15}$$

Where: $\sigma$ is standard deviation of data; $y_i^R$ is the data on response surface.

According to the statistical theory, when the data error is within two times of the standard deviation, the data with reliability probability is 95%; When the data error is within the range of three times standard deviation, the data with reliability probability is 99.7%. For example, if the reliability probability is 99.7%, when $|y_i - y_i^R| \leq 3\sigma$, the data of $y_i$ is accepted; when $|y_i - y_i^R| > 3\sigma$, the data of $y_i$ is rejected. After the data is revised, the loss function based on the response surface is Eq (16):

$$L^R(y, f(x)) = \frac{1}{n+m}\left[\sum_{i=1}^{n}\left(y_i - f(x_i)\right)^2 + \sum_{j=1}^{m}\left(y_j^R - f^R\left(x_j\right)\right)^2\right] \tag{16}$$

Where: $L^R(y, f(x))$ is the loss function added with response surface data; $f^R(x_j)$ is the forecast value of DNN added with response surface data.

## 4.2 The revision of loss function gradient

According to Eqs (9) and (10), the iterative formula of gradient descent can be revised to Eqs (17) and (18):

$$W_{ij} = W_{ij} - \alpha\frac{\partial}{\partial W_{ij}}L^R(y, f(x)) \tag{17}$$

$$b_i = b_i - \alpha\frac{\partial}{\partial b_i}L^R(y, f(x)) \tag{18}$$

Eqs (19) and (20) are used to update the parameters $W$ and $b$ in DNN.

$$W_{ij}^{(l)} = W_{ij}^{(l)} - \alpha\frac{\partial}{\partial W_{ij}^{(l)}}L^R(y, f(x)) \tag{19}$$

$$b_i^{(l)} = b_i^{(l)} - \alpha\frac{\partial}{\partial b_i^{(l)}}L^R(y, f(x)) \tag{20}$$

Where: $l$ is the number of layer; $W_{ij}^{(l)}$ is the weight connecting the $j$th node of $l$th layer to the $i$th node of $(l+1)$th layer, $b_i^{(l)}$ is the bias term of the $i$th node of $(l+1)$th layer.

The goal is to calculate the values of $W$ and $b$ of all layers, where the values of $W$ and $b$ are random values that have been initialized to be close to 0 and $\alpha$ is a preset learning rate. If the partial derivatives $\partial L^R(y, f(x))/\partial W_{ij}^{(l)}$ and $\partial L^R(y, f(x))/\partial b_i(l)$ of each layer are calculated, the values of $W$ and $b$ can be updated according to Eqs (19) and (20).

## 5 The application of MRSF-DNN in prediction of RBAC compressive strength (RBACCS)

### 5.1 Test materials

The test material is shown in Fig 4, wherein the cementitious material is 32.5 grade Portland cement with a density of 3100kg/m³; the fine aggregate is ordinary river sand with a fineness modulus of 2.76, water content of 0.1% and apparent density of 2640 kg/m³; the coarse aggregate is recycled brick aggregate with a moisture content of 2.46%, water absorption of 8.18%, apparent density of 2100 kg/m3 and crushing index of 30.59%.

### 5.2 Test design

In order to verify the applicability of MRSF-DNN model, RBAC compressive strength (RBACCS) test was carried out, in which CA volume content, FA volume content and W/C were taken as three variables. RBA as CA was used in this test plan as shown in Table 1.

In order to prevent the dispersion of test data, three cube with side length of 150mm specimens were made in each mix proportion. When the difference of RBACCS of the three specimens does not exceed 15%, the average value of RBACCS of the three specimens is taken as the RBACCS; when the difference of RBACCS of one of the three specimens exceeds 15%, the data with large error will be discarded, the average value of RBACCS of the other two specimens will be taken as RBACCS; when the difference between RBACCS of two of the three specimens exceeds 15%, RBACCS of the three specimens will be discarded, three new specimens need to be made and tested again.

### 5.3 Analysis of test results

The test result can be found in Fig 5, in which it is shown that the RBACCS shows similar upward and downward trend with different W/C (S1 Table). The RBACCS increases gradually

(a)  (b)  (c)

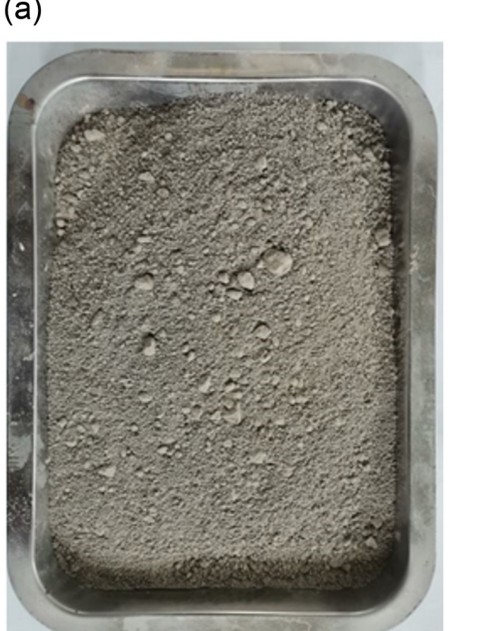 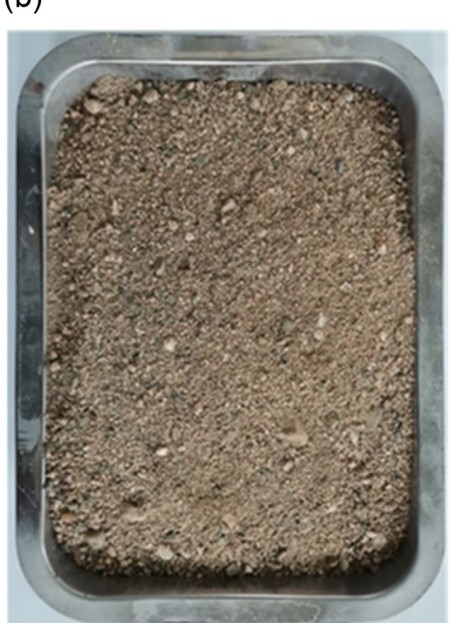 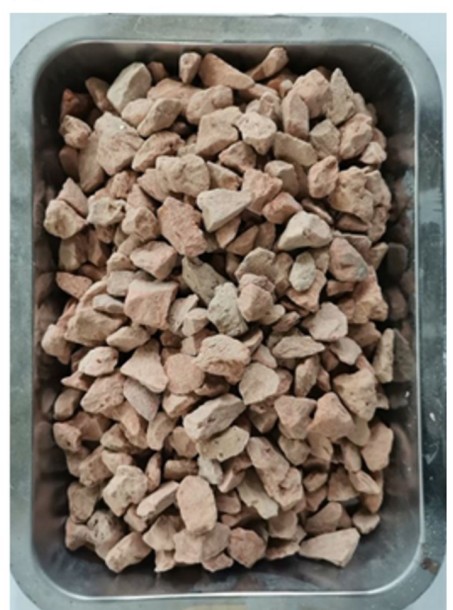

**Fig 4. Test materials: (a) cement (b) sand (c) RBA.**

**Table 1. Mix proportion plan of RBAC.**

| Specimen number | W/C | Content of mix proportion in 1m³ RBAC | | | | | |
|---|---|---|---|---|---|---|---|
| | | Water (kg) | Cement (kg) | RBA (kg) | RBA (m³) | Sand (kg) | Sand (m³) |
| H1a/1b/1c/1d | 0.41/0.46/0.52/0.63 | 278/294/312/339 | 679/640/600/539 | 630 | 0.3 | 475 | 0.18 |
| H2a/2b/2c/2d | 0.41/0.46/0.52/0.63 | 257/272/288/313 | 627/590/554/497 | 630 | 0.3 | 577 | 0.22 |
| H3a/3b/3c/3d | 0.41/0.46/0.52/0.63 | 236/249/264/287 | 574/541/507/456 | 630 | 0.3 | 686 | 0.26 |
| H4a/4b/4c/4d | 0.41/0.46/0.52/0.63 | 252/266/282/307 | 614/578/542/487 | 735 | 0.35 | 475 | 0.18 |
| H5a/5b/5c/5d | 0.41/0.46/0.52/0.63 | 230/243/258/281 | 561/529/496/446 | 735 | 0.35 | 577 | 0.22 |
| H6a/6b/6c/6d | 0.41/0.46/0.52/0.63 | 209/220/234/255 | 509/480/450/404 | 735 | 0.35 | 686 | 0.26 |
| H7a/7b/7c/7d | 0.41/0.46/0.52/0.63 | 225/237/252/274 | 548/517/484/435 | 840 | 0.4 | 475 | 0.18 |
| H8a/8b/8c/8d | 0.41/0.46/0.52/0.63 | 203/215/228/248 | 496/467/438/394 | 840 | 0.4 | 577 | 0.22 |
| H9a/9b/9c/9d | 0.41/0.46/0.52/0.63 | 182/193/204/222 | 444/418/392/352 | 840 | 0.4 | 686 | 0.26 |
| H10a/10b/10c/10d | 0.41/0.46/0.52/0.63 | 208/220/233/253 | 506/477/447/402 | 945 | 0.45 | 475 | 0.18 |
| H11a/11b/11c/11d | 0.41/0.46/0.52/0.63 | 186/205/209/227 | 454/446/401/361 | 945 | 0.45 | 577 | 0.22 |
| H12a/12b/12c/12d | 0.41/0.46/0.52/0.63 | 165/174/185/201 | 402/379/355/319 | 945 | 0.45 | 686 | 0.26 |
| H13a/13b/13c/13d | 0.41/0.46/0.52/0.63 | 171/181/192/209 | 418/394/369/332 | 1050 | 0.5 | 475 | 0.18 |
| H14a/14b/14c/14d | 0.41/0.46/0.52/0.63 | 150/159/168/183 | 366/344/323/290 | 1050 | 0.5 | 577 | 0.22 |
| H15a/15b/15c/15d | 0.41/0.46/0.52/0.63 | 128/136/144/157 | 313/295/277/249 | 1050 | 0.5 | 686 | 0.26 |
| H16a/16b/16c/16d | 0.41/0.46/0.52/0.63 | 145/153/162/176 | 353/332/311/280 | 1155 | 0.55 | 475 | 0.18 |
| H17a/17b/17c/17d | 0.41/0.46/0.52/0.63 | 123/130/138/150 | 300/283/265/238 | 1155 | 0.55 | 577 | 0.22 |
| H18a/18b/18c/18d | 0.41/0.46/0.52/0.63 | 102/107/114/124 | 248/234/219/197 | 1155 | 0.55 | 686 | 0.26 |

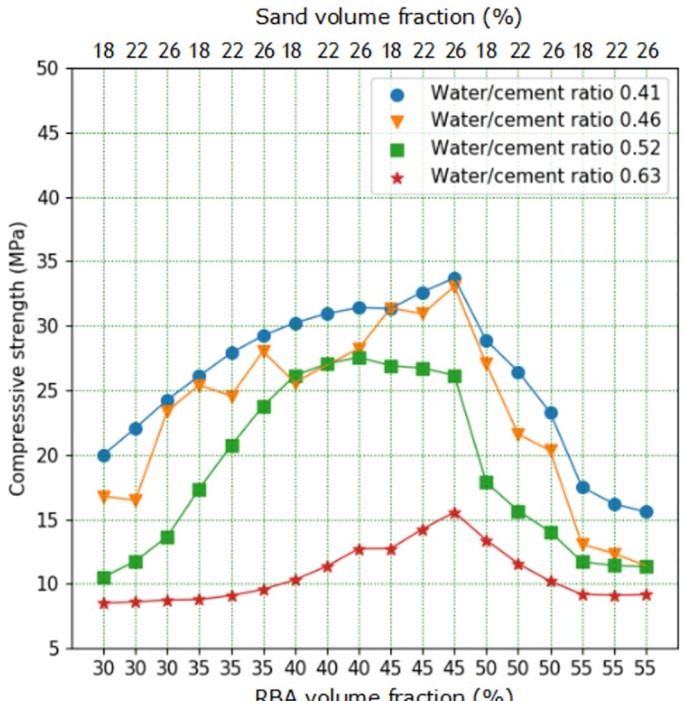

**Fig 5. Influence of volume content of CA and FA on RBACCS.**

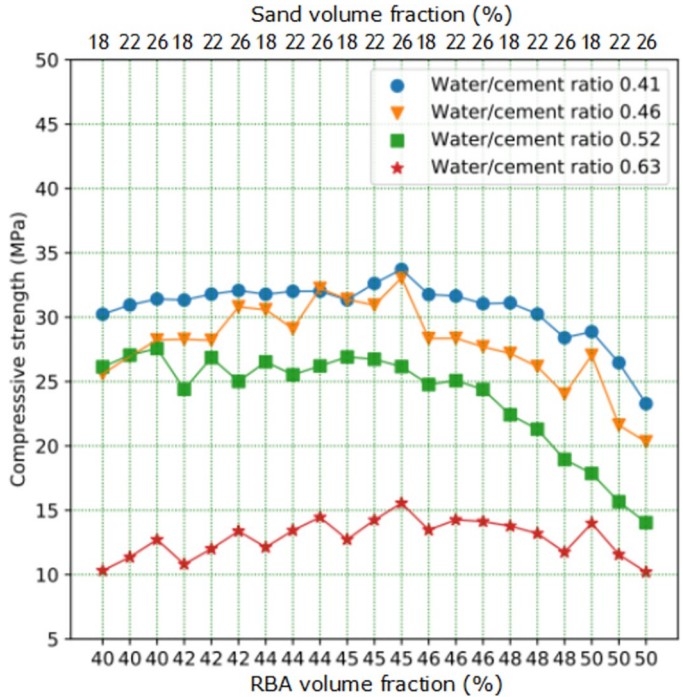

**Fig 6. The RBACCS with RBA volume fraction in the range of 40% ~ 50%.**

with the increase of CA volume content. When CA volume content exceeds about 45%, the RBACCS decreases rapidly. Furthermore, the volume content of CA is the main factor affecting the strength of concrete, while the volume content of FA has no obvious influence. This may be due to the small variation range in the volume content of FA. At the same time, with the increase of the volume content of CA and FA, the volume content of cement mortar decreases, leading to the decrease in bond strength between CA and FA [41, 42].

It can also be found from Fig 5 that when the volume content of CA is between 40% and 50%, higher strength concrete can be obtained. So RBAC cube specimens with CA volume content of 42%, 44%, 46% and 48% respectively and FA volume content of 18%, 22% and 26% respectively were made (S2 Table). The test results can be found in Fig 6, when W/C is 0.63, the volume content of CA has little effect on RBACCS. RBACCS with different volume contents of CA and FA is between 10MPa and 15MPa. When the volume content of CA exceeds about 46%, RBACCS with W/C of 0.41, 0.46 and 0.52 shows a downward trend.

## 5.4 Establishment and analysis of MRSF-DNN model

According to Fig 5, it can be found that RBACCS is not only related to W/C, but also closely related to CA and FA volume content. Moreover, the connection between them shows nonlinear characteristics as shown in Figs 7 and 8. In Figs 7 and 8, $P_g$ is CA volume content, $P_s$ is FA volume content, $m_w/m_c$ is W/C, $f_{cu}$ is RBACCS.

MRSF was used to optimize DNN. Firstly, both the data on the response surface and the test data were used as training samples, the data exceed triple standard deviation on the response surface will be eliminated, which ensures the accuracy of the training samples. Secondly, the test data were expanded by the data on the response surface, which is helpful to the stability of MRSF-CNN prediction.

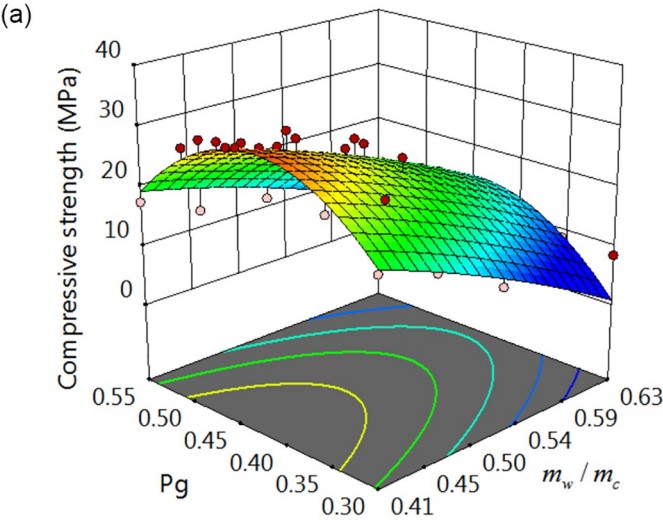

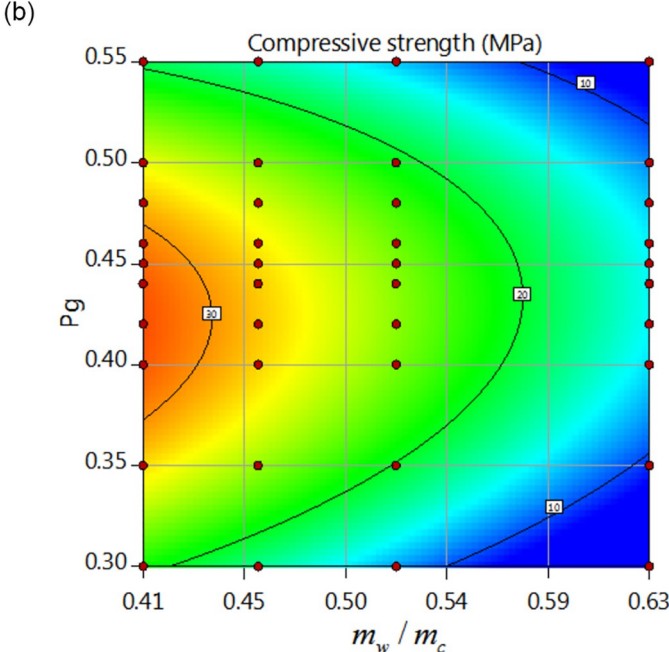

**Fig 7. Response surface of $P_g$, $m_w/m_c$ and $f_{cu}$: (a) Three dimensional response surface, (b) Response surface projection.**

The calculation process of MRSF-DNN is as follows:

① Both the data on the response surface and the test data were used as input data;

② The input data is processed by weight values and bias terms; The output result of hidden layer is mapped nonlinearly by activation function;

③ The data exceed triple standard deviation on the response surface will be eliminated;

④ The errors are fed back to the input layer until the loss function is stable;

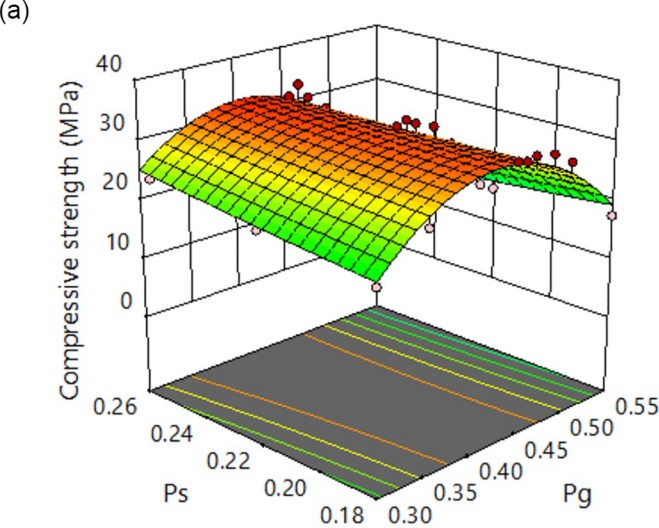

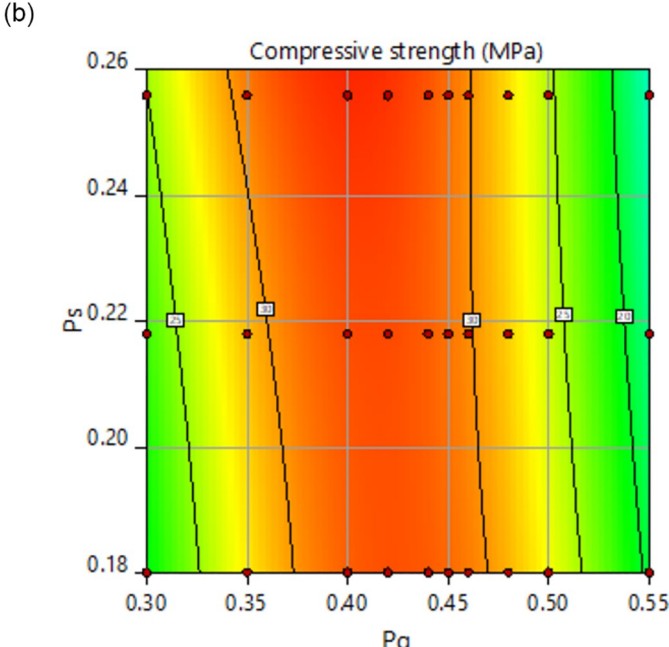

**Fig 8. Response surface of $P_s$, $P_g$ and $f_{cu}$: (a) Three dimensional response surface, (b) Response surface projection.**

⑤ The data after training is output.

In this paper, the input variables include CA volume content, FA volume content and W/C, the output variable is RBACCS. The initial weight values of MRSF-DNN model were randomly generated, the initial bias term was 0. The original response surface was Eq (13). The determination of loss function, the number of hidden layers and the number of calculation steps needs to be calculated.

(1) The determination of loss function

The impact of different activation functions on the loss function is shown in Fig 9, it can be found that the Sigmaid activation function has the best effect. When the calculation step is in the range of 0 to $3 \times 10^5$, the loss function value decreases rapidly; When the calculation step is

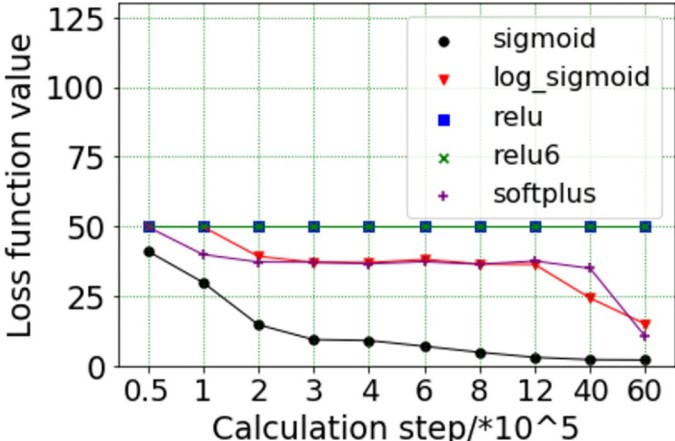

**Fig 9. Influence of activation function on loss function.**

greater than $3\times10^5$, the decline speed of the loss function value slows down; When the calculation step is at about $6\times10^6$, the loss function value is basically stable. The effects of Log_sigmoid and Softplus activation function are similar, when the calculation step is in the range of 0 to $2\times10^5$, the loss function value decreases rapidly; When the calculation step is in the range of $2\times10^5$ to $1.2\times10^6$, the loss function value has almost no change; When the calculation step is greater than $1.2\times10^6$, the loss function value rapidly decreases again and is close to the loss function value of the Sigmaid activation function at $6\times10^6$ calculation step. The effects of Relu and Relu6 activation function are similar, the loss function values of Relu and Relu6 activation function do not converge as the calculation step increases.

(2) The determination of calculation steps

The Sigmoid was selected as the activation function and the loss function value was calculated based on the test results. The results showed that when the calculation step is $4\times10^6$, the loss function value is 2.15; When the calculation step is $6\times10^6$, the loss function value is 2.04; When the calculation step is $1\times10^7$, the loss function value remains 2.04. So when the calculation step is about $6\times10^6$, the loss function value is basically stable. In this paper, the calculation step for the convergence and stability of the loss function is selected as $6\times10^6$.

(3) The determination of the number of hidden layers

The number of hidden layers reflects the times of nonlinear mapping of neural network model to data. When the number of hidden layers are few, the nonlinear mapping effect is not good; When the number of hidden layers are many, it will affect the calculation speed. So it is necessary to find the optimal number of hidden layers. The impact of the number of hidden layers on the loss function is shown in Fig 10. It can be seen that when the number of hidden layers is 5, the loss function values show a large dispersion and are greater than other situations in the calculation steps of $1\times10^5$ to $8\times10^5$. When the number of hidden layers are 10, the loss function values are larger than other situations, although the loss function values are no significant dispersion in the calculation steps of $1\times10^5$ to $6\times10^6$. The calculation results for hidden layers of 20, 30 and 40 are very close. Therefore, considering both computational accuracy and speed, the number of hidden layers of 20 is selected in this paper.

According to the calculation results in part (1), (2) and (3), the activation function was *Sigmoid*(), the amount of hidden layer was 20, the loss function was Eq (16), Gradient Descent Optimizer was selected, the learning steps were set as 6000,000. The structure of MRSF-DNN model is displayed in Fig 11.

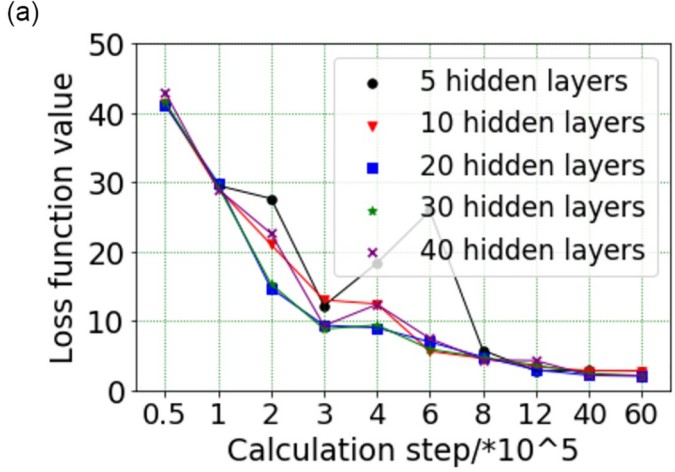

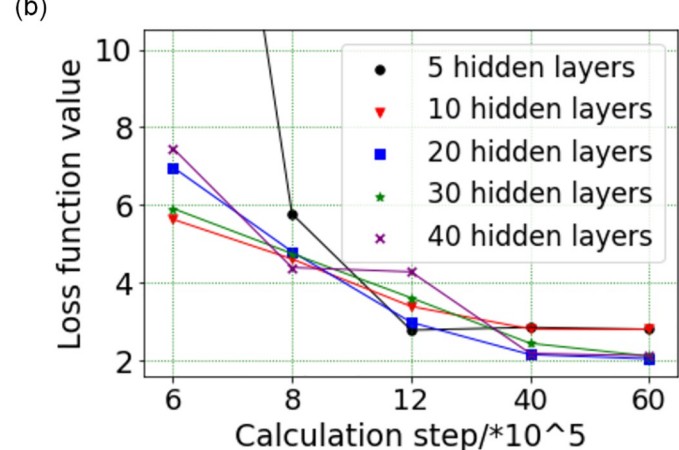

**Fig 10. The effect of the number of hidden layers on the loss function: (a) The calculation steps are $5\times10^4$ to $6\times10^6$ (b) The calculation steps are $6\times10^5$ to $6\times10^6$.**

After 6000,000 steps training, the prediction results of MRSF-DNN and DNN can be found in Fig 12, it is show that the predicted results calculated by MRSF-DNN and DNN are all very close to the real values. After calculation that the related coefficient of MRSF-DNN and DNN between the real values and the forecast values is 0.99 and 0.96, respectively. It is also found from Fig 12(b) and 12(c) that the relative errors between the actual and predicted values of MRSF-DNN ranges from—0.5% to 1%, while the relative errors between the actual and predicted values of DNN ranges from 20% to 15%. So MRSF-DNN shows high prediction accuracy than DNN.

### 5.5 Extended analysis of DNN and MRSF-DNN of RBACCS

In order to compare the forecast effect of DNN and MRSF-DNN, DNN and MRSF-DNN were used in variable parameter extended analysis with CA volume content, FA volume content and W/C as IVs respectively. In the extended analysis with W/C as variable, W/C with 0.43, 0.49 and 0.57 are selected respectively. According to above test, with a constant of CA and FA volume content, the RBACCS decreases with W/C increasing. The prediction results of DNN and MRSF-DNN are displayed in Fig 13(a) and 13(b). It can be found in Fig 13(a) that

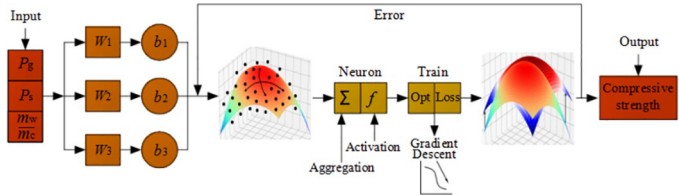

**Fig 11. The RSM-DNN model structure.**

RBACCS with W/C of 0.57 is between that with W/C of 0.63 and 0.52, which conforms to the law of CCS. The law of RBACCS with W/C of 0.43 and 0.49 is similar to that with W/C of 0.57. So DNN can be used to extend analysis of RBACCS. The prediction results of MRSF-DNN are shown in Fig 13(b), which are similar to DNN. So MRSF-DNN also can be used to extend

(a)

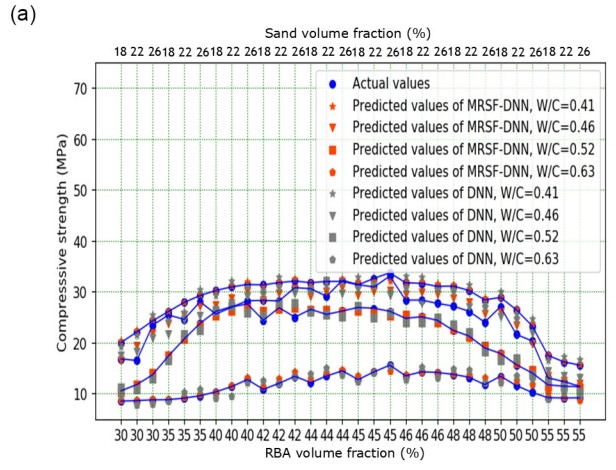

(b)

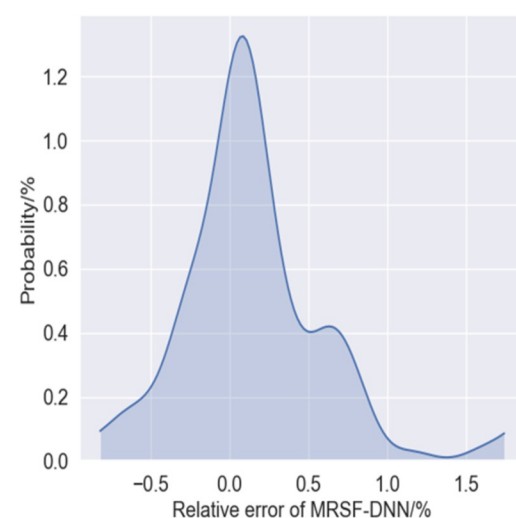

(c)

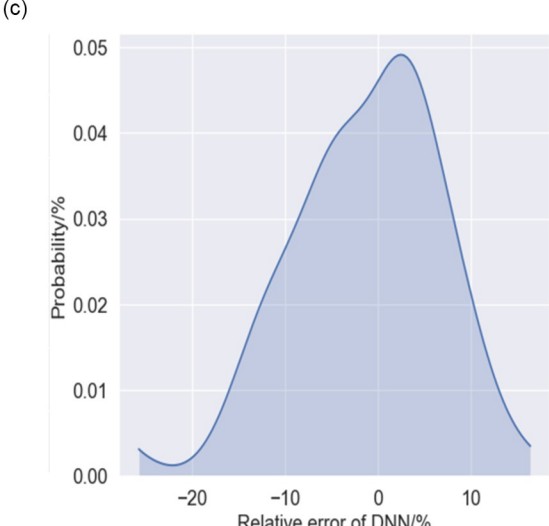

**Fig 12. MRSF-DNN and DNN training results: (a) The training results of MRSF-DNN and DNN, (b) Probability distribution of MRSF-DNN, (c) Probability distribution of DNN.**

(a)

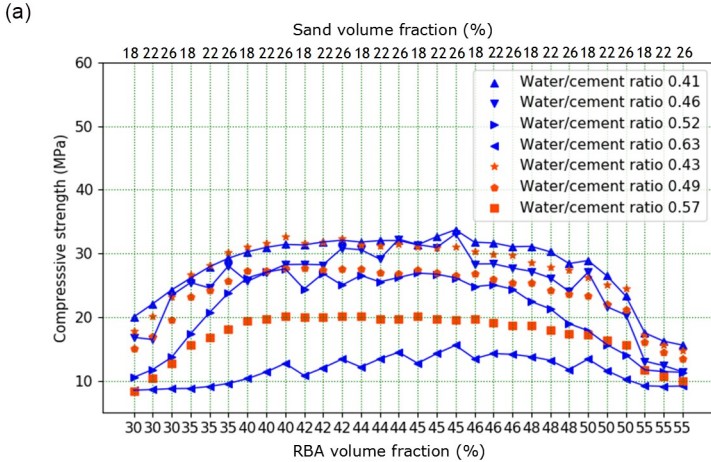

(b)

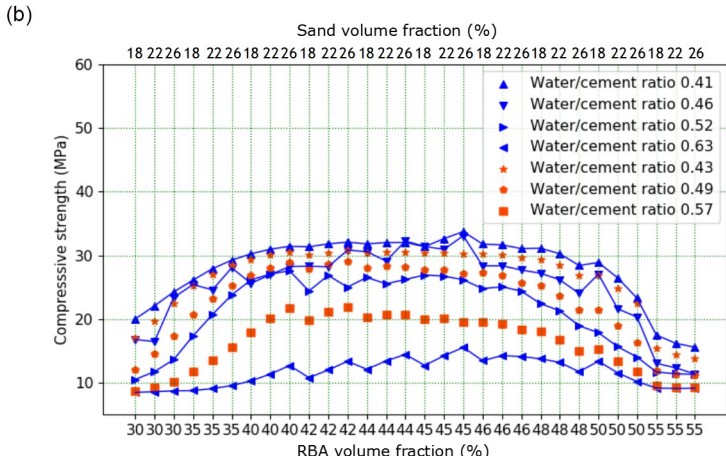

**Fig 13. Extended analysis results of DNN and MRSF-DNN: (a) Extended analysis results of DNN, (b) Extended analysis results of MRSF-DNN.**

analysis of RBACCS. However, comparing the data in Fig 13(a) and 13(b), it is found that MRSF-DNN can change according to the fluctuation of the original data, but DNN does not have this ability. From the data in Fig 13(a), it indicates that the RBACCS with W/C of 0.43 predicted by DNN exceeds that with W/C of 0.41, which does not conform to the law of CCS. However, there are no abnormal values in forecast values of MRSF-DNN in Fig 13(b). So MRSF-DNN has better generalization ability than DNN. This is because the biggest advantage of MRSF-DNN is that the errors of input and training data which exceed σ of the response surface function can be eliminated.

In the extended analysis with CA volume content as variable, CA volume content of 31%, 32%, 33%, 34%, 51%, 52%, 53% and 54% are selected respectively. The prediction results of DNN and MRSF-CNN are displayed in Fig 14(a) and 14(b). According to above test, when CA volume content is between 30% and 35%, the RBACCS show an increasing trend; when CA volume content is between 50% and 55%, the RBAC compressive show a decreasing trend. From Fig 14(a) and 14(b) it is displayed that the RBACS with small W/C is higher than that with large W/C. Under different W/Cs when CA volume content is between 30% and 35%, the RBACCS show an increasing trend; when CA volume content is between 50% and 55%, the RBACCS show a decreasing trend. These are in accordance with the above test results. So

(a)

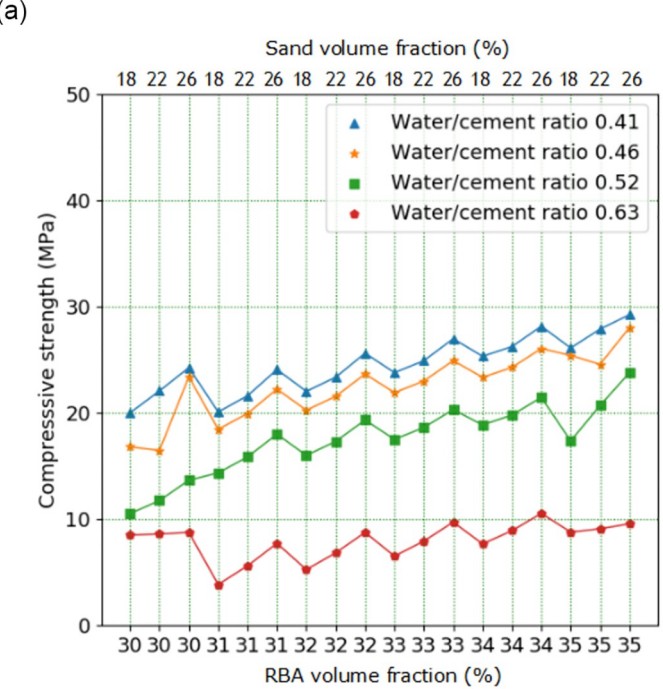

(b)

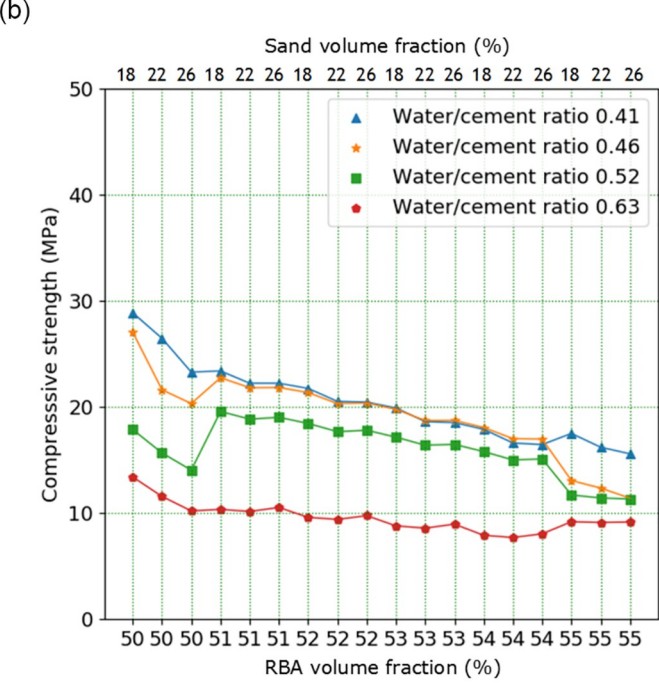

**Fig 14. Extended analysis results of DNN: (a) RBA volume content is in the range of 30%-35%, (b) RBA volume content is in the range of 50%-55%.**

(a)

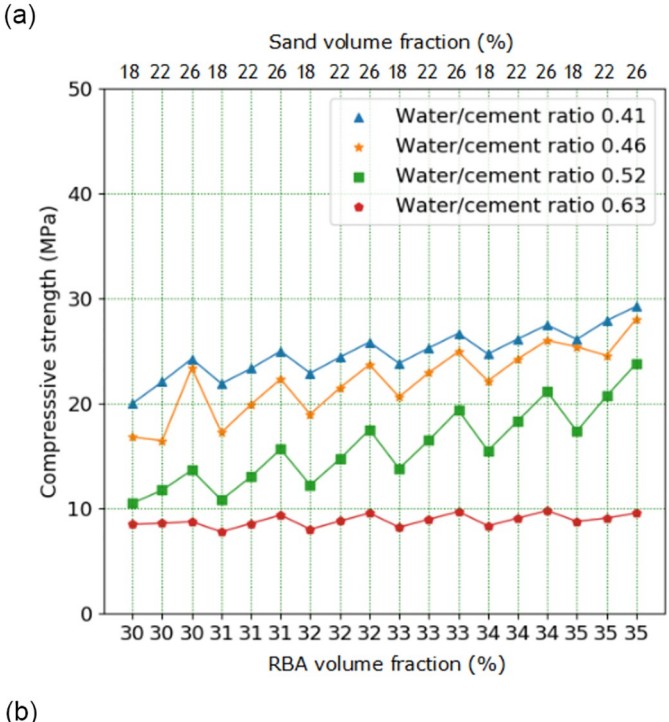

(b)

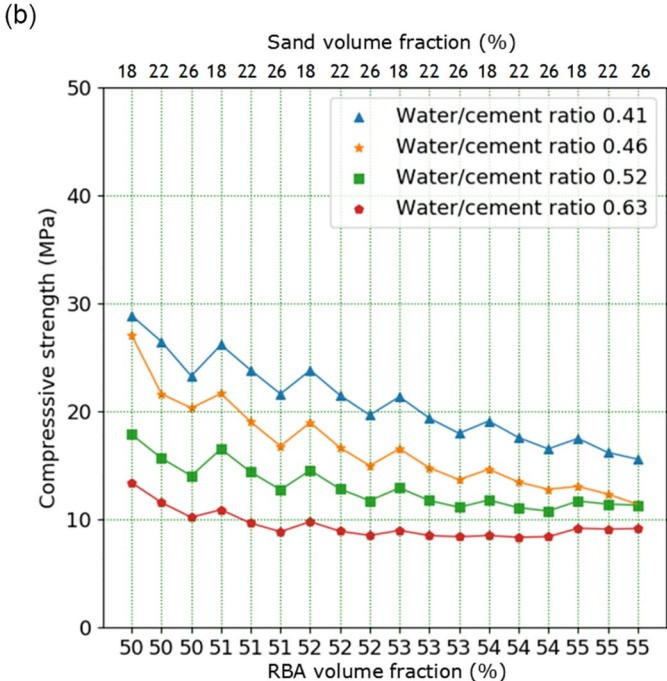

**Fig 15. Extended analysis results of MRSF-DNN: (a) RBA volume content is in the range of 30%-35%, (b) RBA volume content is in the range of 50%-55%.**

DNN can be used to extend analysis of RBACCS. The prediction results of MRSF-DNN are shown in Fig 15(a) and 15(b), which are similar to DNN. So MRSF-DNN also can be used to extend analysis of RBACCS. However, comparing the data in Figs 14(b) and 15(b), it is found that when W/C is in the range of 0.41 to 0.52, the RBACCS predicted by DNN is relatively

close, while the distribution of RBACCS predicted by MRSF-DNN is relatively uniform. So this once again shows that MRSF-DNN has better generalization ability than DNN.

In summary, MRSF-DNN method not only has high accuracy in multivariate concrete strength prediction, but also has good prediction accuracy in extended analysis. The method of MRSF-DNN is not only used to predict the concrete strength of multiple independent variables, but also can be used to predict the mechanical properties and durability of concrete with multiple independent variables. The biggest advantage of MRSF-DNN is that the errors of input and training data which exceed σ of the response surface function can be eliminated to ensure the prediction accuracy and stability.

## 6 Conclusion

In order to improve the accuracy of predicting concrete strength with multiple independent variables, in this paper, MRSF was used to revise input data and training data, the loss function based on the data on the response surface was derived, DNN based on MRSF (MRSF-DNN) was established. The result show that: In prediction analysis, MRSF-DNN shows high prediction accuracy than DNN, the relative errors between the actual and predicted values of MRSF-DNN ranges from—0.5% to 1%, while the relative errors between the actual and predicted values of DNN ranges from 20% to 15%. But in extended analysis, MRSF-DNN has more stable prediction ability and stronger generalization ability than DNN. So the method of MRSF-DNN is not only used to predict the concrete strength with multiple independent variables, but also provides a new idea for the prediction of the mechanical properties and durability of concrete with multiple independent variables.

## Supporting information

**S1 Table. Summary of datasets with RBA volume contents of 30%-55%.** The datasets in the experiment (72). Water/cement ratio, sand volume fraction, and RBA volume fraction in that datasets are considered.
(XLS)

**S2 Table. Summary of datasets with RBA volume contents of 40%-50%.** The datasets in the experiment (84). Water/cement ratio, sand volume fraction, and RBA volume fraction in that datasets are considered.
(XLS)

## Acknowledgments

We acknowledge the experimental equipment provided by Changzhou Institute of Technology. We also thank the anonymous reviewers for their highly constructive suggestions, which helped to improve this manuscript.

## Author Contributions

**Conceptualization:** Xiaohong Chen.

**Data curation:** Yueyue Zhang, Pei Ge.

**Formal analysis:** Pei Ge.

**Methodology:** Xiaohong Chen.

**Supervision:** Yueyue Zhang, Pei Ge.

**Validation:** Yueyue Zhang.

**Writing – original draft:** Pei Ge.

**Writing – review & editing:** Xiaohong Chen.

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
