## [Decision Letter · Decision Letter 0]

1 Mar 2023

PONE-D-23-04065Application of multivariable response surface function revised depth neural network in concrete strength predictionPLOS ONE

Dear Dr. Chen,

Thank you for submitting your manuscript to PLOS ONE. After careful consideration, we feel that it has merit but does not fully meet PLOS ONE’s publication criteria as it currently stands. Therefore, we invite you to submit a revised version of the manuscript that addresses the points raised during the review process.

We look forward to receiving your revised manuscript.

Kind regards,

Jiaolong Ren

Academic Editor

PLOS ONE

Journal Requirements:

"The authors declare that they have no known competing financial interests or personal connections that could have appeared to influence the work reported in this paper."

Reviewers' comments:

Reviewer's Responses to Questions

**Comments to the Author**

1. Is the manuscript technically sound, and do the data support the conclusions?

Reviewer #1: Yes

Reviewer #2: Partly

Reviewer #3: Yes

2. Has the statistical analysis been performed appropriately and rigorously? 

Reviewer #1: Yes

Reviewer #2: N/A

Reviewer #3: Yes

3. Have the authors made all data underlying the findings in their manuscript fully available?

Reviewer #1: Yes

Reviewer #2: Yes

Reviewer #3: Yes

4. Is the manuscript presented in an intelligible fashion and written in standard English?

Reviewer #1: No

Reviewer #2: No

Reviewer #3: Yes

5. Review Comments to the Author

Reviewer #1: The manuscript presents a novel approach to improving the accuracy of deep neural network (DNN) predictions by using a multivariable response surface function (MRSF) to revise input and training data. While the study provides valuable insights into the potential of MRSF to enhance DNN performance, there are several areas that could be improved to make the article more accessible and informative for readers.

1. The language and grammar of the manuscript need a lot of revision and polishing. The title of the article needs to be optimized.

2. The abstract contains a number of abbreviations that may not be familiar to all readers. It is suggested that the authors minimize the use of abbreviations and provide full forms for any that are used.

3. The conclusion is brief and does not provide a comprehensive summary of the study. It is suggested that the authors expand the conclusion section to include a summary of the research methodology, key findings, and implications.

4. The Finacial disclosure part does not mention any funding sources for the study. It is suggested that the authors provide information on funding sources, if any, to ensure transparency and to acknowledge any financial support received.

5. While the MRSF-DNN model was used to predict the compressive strength of recycled brick aggregate concrete, little information on the significance or implications of these predictions is provided. It is suggested that the authors discuss the potential applications of this approach and how it could be used in practice to improve the design and construction of concrete structures.

Reviewer #2: Improvement in the language is required. For example, “MRSF-DNN model of recycled brick aggregate concrete (RBAC) compressive strength was established, in which coarse aggregate (CA) volume content, fine aggregate (FA) volume content and water cement ratio (W/C) as influencing factors.”. Please get an editor and correct the mistakes in the language.

The research gap you identified is the use of multivariable response surface function to revise the input data and training data. It is suggested to provide sufficient background knowledge about the multivariable response surface function usage in the introduction; that may be supported with the literature.

The literature survey is not strong enough. Very similar research has been missed; below are some examples.

Deng, F., He, Y., Zhou, S., Yu, Y., Cheng, H. and Wu, X., 2018. Compressive strength prediction of recycled concrete based on deep learning. Construction and Building Materials, 175, pp.562-569.

Chen, H., Li, X., Wu, Y., Zuo, L., Lu, M. and Zhou, Y., 2022. Compressive strength prediction of high-strength concrete using long short-term memory and machine learning algorithms. Buildings, 12(3), p.302.

Chou, J.S., Tjandrakusuma, S. and Liu, C.Y., 2022. Jellyfish search-optimized deep learning for compressive strength prediction in images of ready-mixed concrete. Computational Intelligence and Neuroscience, 2022.

ANN is a mathematical or computational model. Presenting the theoretical background, you may provide proper sources for the theoretical background. Also, the equation should be supported by the references.

Improve the quality of Figures 9 and 10. In figure 10 (a), what would try to highlight with dotted closed loops?

The results were not bench marked with the existing literature. The results of the MRSF-DNN model showed a high prediction accuracy which is close to the real values. What is the other prediction accuracy of the other methods?

The paper has an abrupt end. Please highlight the contribution and document the recommendations for the future research.

Reviewer #3: *Literature review definitely must be updated.

*The materials are poorly characterized, this section can be expanded

* How have the authors chosen the parameters? Explain.

*Results and discussion: The authors should explain the reason for some trends or conclusions instead of only describing the trends in the data.

6. PLOS authors have the option to publish the peer review history of their article (what does this mean?). If published, this will include your full peer review and any attached files.

Reviewer #1: No

Reviewer #2: No

Reviewer #3: No

---

## [Author Response · Author response to Decision Letter 0]

15 Apr 2023

Dear Editors and Reviewers:

　　Thank you for your letter and for the reviewers’ comments concerning our manuscript entitled “Prediction of concrete strength using response surface function modified depth neural network” (ID: PONE-D-23-04065). Those comments are all valuable and very helpful for revising and improving our paper, as well as the important guiding significance to our researches. We have studied comments carefully and have made correction which we hope meet with approval. The main corrections in the paper and the responds to the reviewer’s comments are as flowing:

Reviewers' comments:

Reviewer #1: 

The manuscript presents a novel approach to improving the accuracy of deep neural network (DNN) predictions by using a multivariable response surface function (MRSF) to revise input and training data. While the study provides valuable insights into the potential of MRSF to enhance DNN performance, there are several areas that could be improved to make the article more accessible and informative for readers.

1. The language and grammar of the manuscript need a lot of revision and polishing. The title of the article needs to be optimized.

Response: We have revised and marked red in the original text.

“Prediction of concrete strength using response surface function modified depth neural network”

2. The abstract contains a number of abbreviations that may not be familiar to all readers. It is suggested that the authors minimize the use of abbreviations and provide full forms for any that are used.

Response: We have revised and marked red in the original text.

ABSTRACT

In order to overcome the discreteness of input data and training data in deep neural network (DNN), the multivariable response surface function was used to revise input data and training data in this paper. The loss function based on the data on the response surface was derived, DNN based on multivariable response surface function (MRSF-DNN) was established. MRSF-DNN model of recycled brick aggregate concrete compressive strength was established, in which coarse aggregate volume content, fine aggregate volume content and water cement ratio are influencing factors. Furthermore, the predictive analysis and extended analysis of MRSF-DNN model were carried out. The results show that: MRSF-DNN model had high prediction accuracy, the correlation coefficient between the real values and the forecast values was 0.9882, the relative error was between -0.5% and 1%. Furthermore, MRSF-DNN had more stable prediction ability and stronger generalization ability than DNN.

3. The conclusion is brief and does not provide a comprehensive summary of the study. It is suggested that the authors expand the conclusion section to include a summary of the research methodology, key findings, and implications.

Response: We have revised and marked red in the original text.

In order to improve the accuracy of predicting concrete strength with multiple independent variables, in this paper, MRSF was used to revise input data and training data, the loss function based on the data on the response surface was derived, DNN based on MRSF (MRSF-DNN) was established. The result show that : In prediction analysis, MRSF-DNN shows high prediction accuracy than DNN, the relative errors between the actual and predicted values of MRSF-DNN ranges from - 0.5% to 1%, while the relative errors between the actual and predicted values of DNN ranges from 20% to 15%. But in extended analysis, MRSF-DNN has more stable prediction ability and stronger generalization ability than DNN. So the method of MRSF-DNN is not only used to predict the concrete strength with multiple independent variables, but also provides a new idea for the prediction of the mechanical properties and durability of concrete with multiple independent variables.

4. The Finacial disclosure part does not mention any funding sources for the study. It is suggested that the authors provide information on funding sources, if any, to ensure transparency and to acknowledge any financial support received.

Response: We have revised and marked red in the original text.

Acknowledgment

Financial support was provided by Henan science and technology research plan project in 2021 (212102310394) and Lanzhou University Open Fund in 2021 (202112).

5. While the MRSF-DNN model was used to predict the compressive strength of recycled brick aggregate concrete, little information on the significance or implications of these predictions is provided. It is suggested that the authors discuss the potential applications of this approach and how it could be used in practice to improve the design and construction of concrete structures.

Response: We have revised and marked red in the original text.

The method of MRSF-DNN is not only used to predict the concrete strength of multiple independent variables, but also can be used to predict the mechanical properties and durability of concrete with multiple independent variables. The biggest advantage of MRSF-DNN is that the errors of input and training data which exceed σ of the response surface function can be eliminated to ensure the prediction accuracy and stability.

Reviewer #2: 

Improvement in the language is required. For example, “MRSF-DNN model of recycled brick aggregate concrete (RBAC) compressive strength was established, in which coarse aggregate (CA) volume content, fine aggregate (FA) volume content and water cement ratio (W/C) as influencing factors.”. Please get an editor and correct the mistakes in the language.

Response: The language of the whole manuscript has been modified.

MRSF-DNN model of recycled brick aggregate concrete (RBAC) compressive strength was established, in which coarse aggregate (CA) volume content, fine aggregate (FA) volume content and water cement ratio (W/C) are influencing factors.

2.The research gap you identified is the use of multivariable response surface function to revise the input data and training data. It is suggested to provide sufficient background knowledge about the multivariable response surface function usage in the introduction; that may be supported with the literature.

Response: We have revised and marked red in the original text.

Response surface methodology has been widely used in concrete performance analysis, optimization and prediction. Application of response surface method in concrete performance analysis, Shi et al. [21] used response surface methodology to analyze the effects of silane rubber and nano SiO2 on the pore structure and mechanical properties of concrete. Adamu et al. [22] used response surface methodology to optimize the effects of the amounts of plastic waste, fly ash and graphene nanotube on the strength and water absorption of concrete. The results showed that the optimal mixing amounts of plastic waste, fly ash and graphene nanoplate were 15.3%, 6.07% and 0.22%, respectively. Hua et al. [23] used the response surface method (RSM) to analyze the influence of Seawater/Potassium Silicate, Potassium Hydroxide/Potassium Chloride, Sodium Laureth Ether Sulfate/Benzalkonium Chloride and Hydrogen Peroxide/Nanocellulose on the density of geopolymer foam concrete. Adamu et al. [24] used response surface methodology to analyze the impact of calcium carbide waste and rice husk ash on the water absorption and permeability of concrete. The results showed that both calcium carbide waste and rice husk ash had a negative impact on the durability of concrete, but rice husk ash had a greater negative impact. Tunc et al. [25] used the response surface method to analyze the effects of water cement ratio, aggregate/cement and Los Angeles abrasion rate on the compressive strength and splitting tensile strength of concrete. Ferdosian et al. [26] used response surface methodology to study the effects of silica fume, ultra-fine fly ash and sand as three main constituents on the workability and compressive strength of ultra-high performance concrete. Application of response surface method in multiobjective optimization, Luo et al. [27] provided a method for optimizing the mix ratio of dune sand concrete based on dune sand/fine aggregate, basalt fiber content, water/ cement and sand/aggregate. The results show that the model established by response surface method is effective and can accurately predict the performance of dune sand concrete. Hamada et al. [28] used response surface methodology to study the effects of nano palm oil fuel ash and palm oil clinker partially replacing cement (0, 15% and 30%) and coarse aggregate (0, 50% and 100%) on the workability and compressive strength of concrete. The results show that response surface methodology has achieved satisfactory results in optimizing the amount of nano palm oil fuel ash and palm oil clinker. When containing 0% palm oil clinker and 15% nano palm oil fuel ash, the compressive strength of concrete is the maximum; When containing 100% palm oil clinker and 30% nano palm oil fuel ash, the compressive strength of concrete is the lowest. Amiri et al. [29] studied the effects of water cement ratio, cement content, gravel content and coal gangue content on the compressive strength and water absorption of concrete. A combination of response surface method and expected function method was used to optimize the relationship between independent variables and response variables for six schemes. Kursuncu et al. [30] used the response surface method and artificial neural network methods to study the effects of waste marble powder and rice husk ash partially replacing fine aggregate and cement on the compressive strength, flexural strength, porosity and thermal conductivity of foam concrete. Zhang et al. [31] designed a Box-Behnken model using the response surface method to study the effects of different amounts of silica fume, fly ash and carbon fiber on the compressive strength and strain sensitivity coefficient of sprayed reactive powder concrete. Siamardi et al. [32] established a model using the response surface method to predict the workability and hardening performance of powder based lightweight self-compacting concrete produced by partially replacing normal weight aggregate with coarse grained lightweight expansive clay aggregate. Shahmansouri et al. [33] used response surface methodology to study the effects of sodium hydroxide concentration, natural zeolite and silica fume on the mechanical properties of geopolymer concrete and obtained the optimal design variables. Application of response surface method in prediction of concrete properties, Hammoudi et al. [34] used response surface method and artificial neural network methods to predict the compressive strength of recycled coarse aggregate concrete. The statistical results show that both response surface method and artificial neural network method are powerful tools for predicting compressive strength. However, the artificial neural network model shows better accuracy. Awolusi et al. [35] used response surface methodology to predict and optimize the impact of limestone powder and steel fiber content on the workability and hardening performance of concrete. The results show that the response surface method has high accuracy in predicting the compressive strength, splitting tensile strength, slump and water absorption of concrete. Gupta et al. [36] established a statistical model for predicting the compressive strength of concrete using the response surface method. The results show that the prediction error of the response surface model is about 3.63%. Application of response surface method in concrete mix design, Zhang et al. [37] used response surface methodology to optimize the ratio among the ideal paste thickness, actual paste thickness and void content of recycled aggregate permeable concrete. Güneyisi et al. [38] also used response surface methodology to optimize the relationship among the three parameters of fly ash, metakaolin and cement in the mix ratio of high-performance concrete.

3.The literature survey is not strong enough. Very similar research has been missed; below are some examples.

Response: We have revised and marked red in the original text.

Deng et al. [15] predicted the impact of water cement ratio, recycled coarse aggregate substitution rate, recycled fine aggregate substitution rate and fly ash substitution rate on the compressive strength of recycled aggregate concrete through convolutional neural network. The results show that compared with traditional neural network models, the prediction model based on deep learning has the advantages of high accuracy, high efficiency, and strong generalization ability. Chou et al. [16] proposed a convolutional neural network based on computer vision for predicting the compressive strength of ready-mixed concrete. The results show that the neural network based on computer vision is superior to the neural network based on numerical data in all evaluation indicators. 

4.ANN is a mathematical or computational model. Presenting the theoretical background, you may provide proper sources for the theoretical background. Also, the equation should be supported by the references.

Response: We have revised and marked red in the original text.

5.Improve the quality of Figures 9 and 10. In figure 10 (a), what would try to highlight with dotted closed loops?

Response: We have revised and marked red in the original text.

6.The results were not bench marked with the existing literature. The results of the MRSF-DNN model showed a high prediction accuracy which is close to the real values. What is the other prediction accuracy of the other methods?

Response: We have revised and marked red in the original text.

After 6000,000 steps training, the prediction results of MRSF-DNN and DNN can be found in Fig.12, it is show that the predicted results calculated by MRSF-DNN and DNN are all very close to the real values. After calculation that the related coefficient of MRSF-DNN and DNN between the real values and the forecast values is 0.99 and 0.96, respectively. It is also found from Fig.12(b) and (c) that the relative errors between the actual and predicted values of MRSF-DNN ranges from - 0.5% to 1%, while the relative errors between the actual and predicted values of DNN ranges from 20% to 15%. So MRSF-DNN shows high prediction accuracy than DNN.

7.The paper has an abrupt end. Please highlight the contribution and document the recommendations for the future research.

Response: We have revised and marked red in the original text.

Reviewer #3: 

Literature review definitely must be updated.

Response: We have revised and marked red in the original text.

2.The materials are poorly characterized, this section can be expanded

Response: We have revised and marked red in the original text.

The test material is shown in Fig.4, wherein the cementitious material is 32.5 grade Portland cement with a density of 3100kg/m3; the fine aggregate is ordinary river sand with a fineness modulus of 2.76, water content of 0.1% and apparent density of 2640 kg/m3; the coarse aggregate is recycled brick aggregate with a moisture content of 2.46%, water absorption of 8.18%, apparent density of 2100 kg/m3 and crushing index of 30.59%.

3.How have the authors chosen the parameters? Explain.

Response: We have revised and marked red in the original text.

　　The initial weight values of MRSF-DNN model were randomly generated, the initial bias term was 0. The original response surface was Eq.(13). The determination of loss function, the number of hidden layers and the number of calculation steps needs to be calculated.

　　（1）The determination of loss function

　　The impact of different activation functions on the loss function is shown in Fig.9, it can be found that the Sigmaid activation function has the best effect. When the calculation step is in the range of 0 to 3×105, the loss function value decreases rapidly; When the calculation step is greater than 3×105, the decline speed of the loss function value slows down; When the calculation step is at about 6×106, the loss function value is basically stable. The effects of Log_sigmoid and Softplus activation function are similar, when the calculation step is in the range of 0 to 2×105, the loss function value decreases rapidly; When the calculation step is in the range of 2×105 to 1.2×106, the loss function value has almost no change; When the calculation step is greater than 1.2×106, the loss function value rapidly decreases again and is close to the loss function value of the Sigmaid activation function at 6×106 calculation step. The effects of Relu and Relu6 activation function are similar, the loss function values of Relu and Relu6 activation function do not converge as the calculation step increases.

　　（2）The determination of calculation steps

　　The Sigmoid was selected as the activation function and the loss function value was calculated based on the test results. The results showed that when the calculation step is 4×106, the loss function value is 2.15; When the calculation step is 6×106, the loss function value is 2.04; When the calculation step is 1×107, the loss function value remains 2.04. So when the calculation step is about 6×106, the loss function value is basically stable. In this paper, the calculation step for the convergence and stability of the loss function is selected as 6×106.

　　（3）The determination of the number of hidden layers

　　The number of hidden layers reflects the times of nonlinear mapping of neural network model to data. When the number of hidden layers are few, the nonlinear mapping effect is not good; When the number of hidden layers are many, it will affect the calculation speed. So it is necessary to find the optimal number of hidden layers. The impact of the number of hidden layers on the loss function is shown in Fig.10. It can be seen that when the number of hidden layers is 5, the loss function values show a large dispersion and are greater than other situations in the calculation steps of 1×105 to 8×105. When the number of hidden layers are 10, the loss function values are larger than other situations, although the loss function values are no significant dispersion in the calculation steps of 1×105 to 6×106. The calculation results for hidden layers of 20, 30 and 40 are very close. Therefore, considering both computational accuracy and speed, the number of hidden layers of 20 is selected in this paper.

　　According to the calculation results in part (1), (2) and (3), the activation function was Sigmoid(), the amount of hidden layer was 20, the loss function was Eq.(16), Gradient Descent Optimizer was selected, the learning steps were set as 6000,000. The structure of MRSF-DNN model is displayed in Fig.11.

4.Results and discussion: The authors should explain the reason for some trends or conclusions instead of only describing the trends in the data.

Response: We have revised and marked red in the original text.

---

## [Decision Letter · Decision Letter 1]

2 May 2023

Prediction of concrete strength using response surface function modified depth neural network

PONE-D-23-04065R1

Dear Dr. Chen,

We’re pleased to inform you that your manuscript has been judged scientifically suitable for publication and will be formally accepted for publication once it meets all outstanding technical requirements.

Kind regards,

Jiaolong Ren

Academic Editor

PLOS ONE

Additional Editor Comments (optional):

Reviewers' comments:

Reviewer's Responses to Questions

**Comments to the Author**

1. If the authors have adequately addressed your comments raised in a previous round of review and you feel that this manuscript is now acceptable for publication, you may indicate that here to bypass the “Comments to the Author” section, enter your conflict of interest statement in the “Confidential to Editor” section, and submit your "Accept" recommendation.

Reviewer #1: All comments have been addressed

Reviewer #2: (No Response)

Reviewer #3: All comments have been addressed

2. Is the manuscript technically sound, and do the data support the conclusions?

Reviewer #1: Yes

Reviewer #2: Yes

Reviewer #3: Yes

3. Has the statistical analysis been performed appropriately and rigorously? 

Reviewer #1: Yes

Reviewer #2: Yes

Reviewer #3: N/A

4. Have the authors made all data underlying the findings in their manuscript fully available?

Reviewer #1: Yes

Reviewer #2: Yes

Reviewer #3: Yes

5. Is the manuscript presented in an intelligible fashion and written in standard English?

Reviewer #1: Yes

Reviewer #2: Yes

Reviewer #3: Yes

6. Review Comments to the Author

Reviewer #1: (No Response)

Reviewer #2: Although responses said those have been added, the sources (references) for the theoretical background cannot be found in the original texts. It is better to have the citations for all equations. Anyway no more reviews are not required.

All other responses for the concerns are at the satisfactory level.

Reviewer #3: (No Response)

7. PLOS authors have the option to publish the peer review history of their article (what does this mean?). If published, this will include your full peer review and any attached files.

Reviewer #1: No

Reviewer #2: No

Reviewer #3: No

---

## [Editor Report · Acceptance letter]

8 May 2023

PONE-D-23-04065R1 

Prediction of concrete strength using response surface function modified depth neural network 

Dear Dr. Chen:

I'm pleased to inform you that your manuscript has been deemed suitable for publication in PLOS ONE. Congratulations! Your manuscript is now with our production department. 

Kind regards, 

on behalf of

Dr. Jiaolong Ren 

Academic Editor

PLOS ONE